# Detection of *Rickettsia* spp. in Animals and Ticks in Midwestern Brazil, Where Human Cases of Rickettsiosis Were Reported

**DOI:** 10.3390/ani13081288

**Published:** 2023-04-09

**Authors:** Lucianne Cardoso Neves, Warley Vieira de Freitas Paula, Luiza Gabriella Ferreira de Paula, Bianca Barbara Fonseca da Silva, Sarah Alves Dias, Brenda Gomes Pereira, Bruno Sérgio Alves Silva, Anaiá da Paixão Sevá, Filipe Dantas-Torres, Marcelo B. Labruna, Felipe da Silva Krawczak

**Affiliations:** 1Veterinary and Animal Science School, Federal University of Goiás, Goiânia 74605-220, Brazil; 2Directorate of Surveillance in Zoonoses, Superintendence of Health Surveillance, Municipal Health Department, Goiânia 74891-135, Brazil; 3Department of Agricultural and Environmental Sciences, Santa Cruz State University, Ilhéus 45662-900, Brazil; 4Laboratory of Immunoparasitology, Department of Immunology, Aggeu Magalhães Institute, Oswaldo Cruz Foundation (Fiocruz), Recife 50740-465, Brazil; 5Department of Preventive Veterinary Medicine and Animal Health, Faculty of Veterinary Medicine and Animal Science, University of São Paulo, São Paulo 05508-270, Brazil

**Keywords:** *Rickettsia bellii*, IFA, PCR, Goiânia, Brazilian Cerrado, tick-borne disease

## Abstract

**Simple Summary:**

Human cases of Brazilian spotted fever (BSF), diagnosed serologically by immunofluorescence assay, have recently been reported in the Goiás state, midwestern Brazil. Because serological cross-reactions among different rickettsial species that belong to the spotted fever group (SFG) are common, the agent responsible for the BSF cases in Goiás remains unknown. We evaluated the presence of anti-*Rickettsia* spp. antibodies in dogs, horses and capybaras (*Hydrochoerus hydrochaeris*), as well as rickettsial DNA in ticks collected from these animals and from the environment, in an area where BSF cases have been reported and two areas under surveillance in Goiás. The DNA of *Rickettsia* that did not belong to the SFG was detected in *Amblyomma dubitatum*, which was identified by DNA sequencing as *Rickettsia bellii*. Seroreactivity to SFG and *R. bellii* antigens was detected in dogs, horses and capybaras, with higher titers for *R. bellii* in dogs and capybaras. These data demonstrate the circulation of SFG rickettsiae in the region and the need for further research to definitively determine the agent responsible for rickettsiosis cases in this area.

**Abstract:**

Brazilian spotted fever (BSF) is the most important tick-borne diseases affecting humans in Brazil. Cases of BSF have recently been reported in the Goiás state, midwestern Brazil. All cases have been confirmed by reference laboratories by seroconversion to *Rickettsia rickettsii* antigens. Because serological cross-reactions among different rickettsial species that belong to the spotted fever group (SFG) are common, the agent responsible for BSF cases in Goiás remains unknown. From March 2020 to April 2022, ticks and plasma were collected from dogs, horses and capybaras (*Hydrochoerus hydrochaeris*), and from the vegetation in an area where BSF cases have been reported and two areas under epidemiological surveillance in Goiás. Horses were infested by *Amblyomma sculptum*, *Dermacentor nitens* and *Rhipicephalus microplus*; dogs by *Rhipicephalus sanguineus* sensu lato (s.l.), *Amblyomma ovale* and *A. sculptum*, and capybaras by *A. sculptum* and *Amblyomma dubitatum*. Adults of *A. sculptum*, *A. dubitatum*, *Amblyomma rotundatum* and immature stages of *A. sculptum* and *A. dubitatum*, and *Amblyomma* spp. were collected from the vegetation. DNA of *Rickettsia* that did not belong to the SFG was detected in *A. dubitatum*, which was identified by DNA sequencing as *Rickettsia bellii*. Seroreactivity to SFG and *Rickettsia bellii* antigens was detected in 25.4% (42/165) of dogs, 22.7% (10/44) of horses and 41.2% (7/17) of capybaras, with higher titers for *R. bellii* in dogs and capybaras. The seropositivity of animals to SFG *Rickettsia* spp. antigens demonstrates the circulation of SFG rickettsiae in the region. Further research is needed to fully determine the agent responsible for rickettsiosis cases in this area.

## 1. Introduction

Among the rickettsiae circulating in Brazil, *Rickettsia rickettsii* and *Rickettsia parkeri* are the most important species from a public health perspective. *R. rickettsii* is the agent of Brazilian spotted fever (BSF), the most severe form of spotted fever and an important zoonosis with a high fatality rate in humans (≥50%), mainly in the southeast of the country [1,2,3]. *R. parkeri* is the causative agent of a milder spotted fever, which has not been related to fatal cases so far, but it is an emerging pathogen in different parts of the country [4,5,6,7,8]. Among other species circulating in Brazil, *Rickettsia bellii* has been detected in more than 25 tick species and is considered to be non-pathogenic for animals and humans [9,10]. However, it has been argued that *R. bellii* could play a role in the ecology and epidemiology of spotted fever group (SFG) rickettsiae by inhibiting their vertical transmission (transovarian transmission) to new generations of ticks [10,11].

Sampling domestic animals and capybaras has been useful for detecting the circulation of SFG rickettsiae in areas where *Amblyomma sculptum*, *Amblyomma aureolatum* and *Amblyomma ovale* are present [1,12,13,14,15,16,17]. As an example, dogs and horses are hosts of tick vectors of SFG rickettsiae in Brazil and are also capable of producing antibodies against *Rickettsia* spp., which are detectable by indirect immunofluorescence assays (IFA). Thus, these animals are reliable sentinels to assess the epidemiological situation of BSF in a given area [12,13,14,15,16,18].

Capybaras also play a crucial role in the epidemiology of BSF. In addition to being sentinels, they act as amplifying hosts of *R. rickettsii*, infecting new populations of *A. sculptum* [19,20]. Although BSF cases have been confirmed in the state of Goiás (including in the Goiânia municipality), our current knowledge about the vectors, their respective hosts and the epidemiology of the disease in this region is still incipient [21,22]. Therefore, the present study aimed to investigate the presence of antibodies to *Rickettsia* spp. in dogs, horses and capybaras, in addition to detecting *Rickettsia* spp. DNA in ticks collected from these animals and from the environment in different areas of Goiânia, Goiás state, midwestern Brazil.

## 2. Materials and Methods

### 2.1. Study Area

This study was performed from March 2020 to April 2022 in the municipality of Goiânia, the capital of the Goiás state, which is located in midwestern Brazil. From an ecological viewpoint, 70% of the state’s territory is part of the Cerrado biome, a tropical savanna ecoregion with the following two distinct seasons: the rainy season (from October to April) and the dry season (from May to September) [23].

Field activities were carried out in an area with a confirmed case of BSF (site A) and two areas (sites B and C) considered by the Superintendência de Vigilância em Saúde de Goiás (SUVISA-GO) as areas under epidemiological surveillance (Figure 1).

The study area at each chosen site was delimited by tracing a radius of 3 km from the place where ticks were collected from the environment (Figure 1). Site A corresponds to the school farm of the Veterinary and Animal Science School (EVZ) of the Federal University of Goiás (UFG) and to the neighborhoods close to the EVZ (16°35′37.3″ S, 49°16′53.5″ W, altitude 718 m); site B corresponds to the Vila Morais neighborhood and the set of neighboring residential neighborhoods, near the Meia Ponte river (16°39′31″ S, 49°13′8″ W, altitude 717 m); and site C corresponds to the peri-urban and rural areas close to the dam of the Ribeirão João Leite reservoir, which is located within the Altamiro de Moura Pacheco State Park (PEAMP), an integral protection conservation unit (16°34′09.2″ S, 49°12′51.2″ W, altitude 742 m).

This study was previously approved by the Chico Mendes Institute for biodiversity (ICMBio Permit No. 70679-5) and by the Institutional Animal Care and Use Committee (CEUA/UFG) of the Federal University of Goiás (protocol 092/19).

### 2.2. Animal Sampling

The sample size for dogs was calculated using an expected frequency of 10% for the detection of antibodies against SFG *Rickettsia* spp. [14,24]. Due to the lack of knowledge of the size of the population of dogs in the study area, an infinite population was considered for the sample calculation, with a margin of error of 5% and a confidence level of 95%, using the EpiInfo^®^ program. The resulting calculated sample size was at least 138 dogs.

For horses and capybaras, we used a convenience sampling strategy because some horse owners were reluctant to participate in the research and due to the inherent difficulties associated with capturing and handling capybaras. Capybaras were drawn into a trap in site A of approximately 90 m^2^, using corn, corn silage and banana leaves as bait. The baits were placed in the afternoon between 4:30 p.m. and 5:30 p.m., and capybaras were captured during the night, with the aid of a net catcher. Once trapped and physically restrained with the net, capybaras were anesthetized with an intramuscular injection of ketamine (10 mg/kg) plus xylazine (0.5 mg/kg). Capybaras were identified with a subcutaneous microchip (Allflex), clinically monitored during the procedure until recovery from anesthesia, and released at the same capture site, as suggested by Neves et al. [25].

### 2.3. Blood Sampling and Tick Collection

Blood samples were collected in tubes with EDTA by venipuncture of the cephalic vein for dogs, jugular vein for horses and saphenous vein for capybaras. The samples were centrifuged at 5000× *g* for 10 min, and the separated plasma was kept at −20 °C until processing [14,26].

Each sampled animal was carefully inspected for the presence of ticks, for 3 min for dogs and capybaras, and 5 min for horses. During the morning, ticks were collected from the environment by flagging on the vegetation [27], covering an area of approximately 120 m^2^ (30 × 40 m) on each site. For this, two white flannels were used, with a pair of collectors for each flannel. Flagging was carried out twice for 1:30 h in each study area and, during this period, flannels were examined every 5–10 m for the presence of ticks. Ticks were removed with the aid of toothless tweezers, placed in 15 mL conical tubes containing isopropyl alcohol and kept at room temperature until taxonomic identification in the laboratory.

Ticks were identified to the species level under a stereomicroscope using descriptions and taxonomic keys [28,29,30,31]. Because there is no taxonomic key for Brazilian *Amblyomma* larvae, they were identified to the genus level only [32].

### 2.4. Detection of Antibodies to Rickettsia spp.

Plasma from dogs, horses and capybaras were tested by indirect immunofluorescence assays (IFA) using crude antigens derived from the following four *Rickettsia* spp. isolates from Brazil: *R. rickettsii* (strain Pampulha) [33], *R. parkeri* (strain Atlantic rainforest) [6], *R. bellii* (strain Mogi) [34] and *R. amblyommatis* (strain Ac37) [35], as previously described by Labruna et al. [36]. Slides with crude antigens were produced as described by Horta et al. [15]. Plasma samples were tested individually using the methodology described by Horta et al. [15]. However, unlike Horta et al. [15], we used plasma instead of serum to detect IgG antibodies. Previous studies showed that there was no difference between serum and plasma samples for antibody detection [37,38]. Briefly, the plasma was diluted in two-fold increments with phosphate-buffered saline (PBS), starting from the 1:64 dilution. Slides were incubated with rabbit anti-dog IgG (Sigma, St Louis, MO, USA), rabbit anti-horse IgG (Sigma, St Louis, MO, USA) and sheep anti-capybara IgG (CCZ, São Paulo, Brazil), coupled with fluorescein isothiocyanate at the 1:1000 dilution for dogs and horses, and at the 1:500 dilution for capybaras. Plasma samples reacting at the 1:64 dilution were titrated at two-fold increments to determine the endpoint titer to each of the four *Rickettsia* antigens. For plasma showing antibody titers to a given *Rickettsia* species at least four-fold higher than those observed for the other *Rickettsia* species, the reaction was considered to be possibly homologous to that *Rickettsia* species or to a closely related species [15,36]. In each slide, a serum previously shown to be non-reactive (negative control) and a known reactive serum (positive control), from the studies of Piranda et al. [39], Ueno et al. [40] and Ramírez-Hernández et al. [20], were tested at the 1:64 dilution.

### 2.5. Molecular Detection of Rickettsia spp. in Ticks and Blood Samples

Of the 5804 ticks collected in this study, a sample of 524 individuals (i.e., 377 *A. sculptum* (72, 98 and 207 from horses, capybaras and vegetation, respectively), 27 *A. dubitatum* (26 and 1 from capybaras and vegetation, respectively), 67 *D. nitens* from horses, and 53 *R. sanguineus* s.l. from dogs) were randomly selected and individually processed for DNA extraction using the guanidine isothiocyanate and phenol/chloroform technique [14] for adult ticks and the boiling extraction method for nymphs [41]. In addition to ticks, 17 blood samples collected from capybaras were also subjected to DNA extraction using the DNAeasy Blood and Tissue Kit (Qiagen, Valencia, CA, USA), following the manufacturer’s recommendations. PCR with the blood of dogs and horses was not performed because previous studies showed the very low sensitivity of this technique for the detection of rickettsial DNA in the blood of dogs and horses that have been experimentally infected with *R. rickettsii* [39,40].

DNA from ticks and capybara blood samples were tested by a TaqMan real-time qPCR assay that targeted a 147-bp fragment of the rickettsial citrate synthase (*gltA*) gene [42,43,44]. The qPCR-positive samples were tested by the following two conventional PCR assays: one using primers CS-78 and CS-323 for the *gltA* gene [42], and another using primers Rr190.70p and Rr190.602n targeting the 190 kDa outer membrane protein (*ompA*) gene of SFG rickettsiae [45]. Negative samples were further tested using PCR protocols targeting the *16S rDNA* gene of ticks [46] or the cytochrome *b* (*cytB*) gene of mammals [47] in order to validate the DNA extraction protocol. If a sample did not produce any product in these PCR assays, the sample was discarded from the study.

PCR products for the *gltA*, *ompA*, *16S rDNA* and *cytB* genes were stained with SYBR Safe (Invitrogen, Carlsbad, CA, EUA), according to the manufacturer’s recommendations, and visualized by electrophoresis in a 1.5% agarose gel using an ultraviolet transilluminator. The *gltA* PCR products were sequenced and the obtained sequences were subjected to BLAST analyses (www.ncbi.nlm.nih.gov/blast; accesed on 20 October 2022) to infer the closest similarities available in GenBank.

### 2.6. Data Analyses

The positivity to antibodies against *Rickettsia* spp. was compared for each animal species and sample sites using a chi-square or Fisher’s exact test with Bonferroni correction. Differences were considered to be significant when *p* < 0.05 and all the analyses were performed in the program R (version 4.2.0), with *packages stats* and *rcompanion*. Ninety-five (95%) confidence intervals (CI) were calculated for prevalence data.

## 3. Results

### 3.1. Reactivity of Antibodies to Rickettsia spp. by IFA

Overall, 25.4% (42/165; 95% CI: 18.8–32.1%) of dogs, 22.7% (10/44; 95% CI: 10.3–35.1%) of horses and 41.2% (7/17; 95% CI: 17.8–64.6%) of capybaras sampled in this study had antibodies (titer ≥ 64) for at least one of the four *Rickettsia* antigens tested herein (Table 1). Of the 42 positive canine plasma, 20 (47.6%), 5 (11.9%), 4 (9.5%) and 1 (2.4%) showed possible homologous reactions to *R. bellii*, *R. rickettsii*, *R. parkeri* and *R. amblyommatis*, respectively. Regarding the horses, 40% (4/10) and 10% (1/10) of the seroreactive animals showed possible homologous reactions to *R. rickettsii* and *R. bellii*, respectively. Among the capybaras, 57.1% (4/7) of the reactive plasma showed possible homologous reactions to *R. bellii* (Table 1).

At site A, the area with a confirmed case for BSF, 57.1% of the seropositive capybaras showed possible homologous reactions to *R. bellii*, with titers ranging from 64 to 512. Among dogs, 16.7% (2/12) and 66.7% (8/12) showed possible homologous reactions to *R. parkeri* and *R. bellii*, respectively, 8.3% (1/12) reacted to *R. rickettsii* and 8.3% (1/12) to *R. amblyommatis*. Possible homologous reactions to *R. rickettsii* and *R. bellii* were detected in 33.3% (3/9) and 11.1% (1/9), respectively, of horses (Table 1).

In places under epidemiological surveillance for BSF (site B), 13.3% (2/15) and 40% (6/15) of the dogs showed possible homologous reactions to *R. rickettsii* and *R. bellii*, respectively. Among the horses, only 9.1% (1/11) of them showed antibodies to SFG rickettsiae, demonstrating possible homologous reactions to *R. rickettsii*. At site C, 13.3% (2/15), 13.3% (2/15) and 40% (6/15) of the dogs demonstrated possible homologous reactions to *R. rickettsii*, *R. parkeri* and *R. bellii*, respectively. In this area, none of the 11 horses evaluated by IFA showed antibodies to *Rickettsia* spp.

Although the IFA positivity for all animals showed different absolute values at site A (31.5%; 28/89), as compared to other sites (22.9% (16/70) and 22.4% (15/67) in site B and C, respectively), there was no significant difference (*X^2^* = 2.186; *df* = 2; *p* = 0.335). The positivity in horses was higher at site A as compared to other sites (Table 1), but statistically significant only between sites A and C (Fisher’s exact test, OR = infinite; *p* = 0.045). The two-tailed *p* value between the sites A and B was 0.164 and between B and C, it was 1.000. Regarding the dogs, the positivity was similar across all sites and there was no statistical difference between them (*X^2^* = 0.116; *df* = 2; *p* = 0.944).

### 3.2. Tick Identification

The percentage of infested animals and the tick species found according to the study site are detailed in Table 2. Overall, 54.5% (90/165; 95% IC: 46.6–61.8%) of the analyzed dogs were infested with ticks. Out of 620 tick specimens collected from dogs, 570 (91.9%) were identified as adults of *R. sanguineus* s.l., 49 (7.9%) as *Amblyomma sculptum* nymphs, and 1 (0.1%) as a female of *Amblyomma ovale*.

Among the horses, 70.4% (31/44; 95% IC: 57.0–83.9%) of the animals were infested by ticks (*n* = 425); 73.8% (314/425) of the ticks were adults of *Dermacentor nitens*, 23.3% (99/425) were adults and 0.2% (1/425) were nymphs of *A. sculptum*, 2.4% (10/425) were adults of *Rhipicephalus microplus*, and 0.2% (1/425) were *Amblyomma* spp. larvae. All capybaras (17/17) were parasitized by ticks (*n* = 785). In particular, 83.6% (656/785) were *A. sculptum* (200 adults and 456 nymphs), 16.3% (128/785) were *Amblyomma dubitatum* (83 adults and 45 nymphs) and 1 was *Amblyomma* sp. larva (Table 2).

A total of 3974 ticks were collected from the environment (one location for each of the sites A, B and C), of which 68.3% (2713) were *Amblyomma* spp. larvae, 31.6% were *A. sculptum* (214 adults and 1043 nymphs), 0.08% were *A. dubitatum* (two adults and one nymph) and one (0.02%) was a female of *Amblyomma rotundatum* (Table 2).

The following voucher tick specimens were deposited in the tick collection ‘Coleção Nacional de Carrapatos do Cerrado’ (CNCC) of the Veterinary and Animal Science School, Federal University of Goiás (accession numbers in parentheses): 100 *Amblyomma* spp. larvae (CNCC 010), 10 *A. sculptum* nymphs (CNCC 011), five males and five females of *A. sculptum* (CNCC 012), two males and two females of *R. microplus* (CNCC 014), five males and five females of *D. nitens* (CNCC 016), one female of *A. ovale* (CNCC 017), one female of *A. rotundatum* (CNCC 018) and five males and five females of *R. sanguineus* s.l. (CNCC 019).

### 3.3. Molecular Detection of Rickettsia spp. in Ticks

Out of 524 tested ticks (Table 3), only 0.4% (2/524) were positive for the rickettsial *gltA* gene upon qPCR testing. These two samples were from *A. dubitatum*, thus corresponding to a positivity of 7.7% in this tick species (2/26) (Table 3). However, none of the samples tested positive for the *ompA* gene. A PCR product was successfully sequenced from the one *A. dubitatum* sample and the corresponding *gltA* gene fragment (350 bp) was 100% identical to *R. bellii*. The rickettsial sequence generated in the present study was deposited in GenBank under the accession number OP718791.

All 17 capybaras were negative for the presence of *Rickettsia* spp. All PCR negative results for *Rickettsia* spp. were confirmed by positive amplification of the *16S rDNA* or *cytB* genes from these ticks or capybara blood samples, respectively.

## 4. Discussion

Our results confirmed the exposure of dogs, horses and capybaras to *Rickettsia* spp. with regard to dogs, the IFA positivity (range: 24–26.8%) was lower than that reported in BSF-endemic areas in the southeastern and southern regions of Brazil, which is around 66% [14,15,18]. In our study areas, we found dogs with antibodies to SFG rickettsiae and parasitized by ticks that can act as vectors of SFG rickettsiae to humans in Brazil [7,48,49]. Nevertheless, in all three study sites, most of the dogs presented serological reactions with higher titers to *R. bellii* than to SFG rickettsiae. These data highlight the presence of *R. bellii*, although the circulation of SFG rickettsiae cannot be ruled out because the lower endpoint titers to SFG could be related to other rickettsial antigens not tested in this study [39]. Regardless, our serological results do not allow us to confirm the circulation of SFG rickettsiae in the study sites.

Our results agree with Neves et al. [50] who carried out a serological survey with dogs from the south and central regions of Goiás. Using the very same rickettsial antigens, they detected an overall positivity of 19%, with 73% of the seropositive dogs showing a homologous reaction to *R. bellii*. The highest frequency of homologous reactions to *R. bellii* among seropositive dogs in our study can be attributed to the fact that this rickettsia frequently infects different species of ticks in Brazil, including *A. ovale* [51,52], a tick species found on dogs in one of the study sites.

Regarding horses, we found that the IFA positivity to *Rickettsia* spp. was higher in the area with a BSF-confirmed case (site A) than in the area under surveillance for rickettsioses (site C). However, only three horses from site A showed possible homologous reactions to *R. rickettsii* and with low titers (128–256), which does not confirm the circulation of *R. rickettsii* in site A, as the titers were relatively low and there are no studies that detected this rickettsia in ticks in site A. Our results were lower than the rates of 77.3% and 44.5% reported by Horta et al. [15] and Souza et al. [53], respectively, in BSF-endemic areas in the southeast region of Brazil. However, in our study, *A. sculptum* was the second most common tick species (23.5%) in horses. These animals can sustain a high density of *A. sculptum* ticks [54] and, together with other factors such as the presence of capybaras and riparian forests, indicate that the sampled areas present factors that are favorable to an increased risk of *R. rickettsii* transmission and, consequently, the occurrence of BSF cases [13,55].

The positivity value of 41.2% detected in capybaras was lower than that found by Pacheco et al. [56] (74%) in non-endemic areas in the southeastern region of Brazil, where a greater number of capybaras with possible homologous reactions to *R. bellii* and with *A. dubitatum* predominating over *A. sculptum* were observed. In our study, 57.1% (4/7) of the capybaras showed a possible homologous reaction *to R. bellii*, suggesting that these animals have been exposed to this bacterium [17,56]. However, *A. sculptum* was the predominant tick species (83.6%) in capybaras compared to *A. dubitatum* (16.3%). It is likely that the high population density of capybaras and the presence of horses at the study site (site A) directly influenced the population size of this tick in the environment [57] and, consequently, on capybaras.

In the three sampled areas, in addition to the presence of horses and capybaras, other factors also contributed to *A. sculptum* being the predominant tick species in the vegetation. These factors include the greater adaptability of *A. sculptum* to anthropized areas [17] and the characteristics of the study sites, which presented a dense vegetation cover with the predominance of degraded pastures, grasses and shrubs. These characteristics provide an adequate microclimate for tick survival [1,58]. Similar to what was observed in our study, de Paula et al. [27] studied the seasonal dynamics of *A. sculptum* at site A and confirmed the predominance of *A. sculptum* over *A. dubitatum* in the environment. The latter species is more abundant in areas that are seasonally flooded [58,59,60], which is not the case of our study sites. Only one specimen of *A. rotundatum* was sampled during the collections in the vegetation (site C). This species has amphibians and reptiles as the main hosts [61] and, although its collection on vegetation has been previously reported, this is unusual due to the fact that *A. rotundatum* displays hunter behavior, thus making its collection difficult by vegetation collection using the cloth dragging technique [62].

Although we detected rickettsial DNA (not belonging to the SFG) in only 0.4% (2/524) of the ticks, we cannot exclude the possible presence of SFG rickettsiae in the study sites. Indeed, less than 1% of *A. sculptum* [49,63] and about 10% of *A. ovale* [16,64,65] ticks tested in previous studies have been found to be positive to *R. rickettsii* and *R. parkeri*, respectively, in areas where human rickettsiosis is endemic in Brazil. Furthermore, recent works have demonstrated the presence of *R. amblyommatis* and *R. felis* in *A. sculptum* and *R. parkeri* in *A. ovale* in Goiás [66,67], thus confirming the circulation of SFG rickettsiae in this state.

In Brazil, *R. bellii* has been identified in different populations of *A. dubitatum* [11,42,68], a finding corroborated by the detection of *R. bellii* DNA in *A. dubitatum* collected from one of the capybaras sampled in the current study. In a similar fashion, the high frequency of possible homologous reactions to *R. bellii* in capybaras tested herein further suggests the circulation of this bacterium in the study areas.

Luz et al. [17] demonstrated that the detection of antibodies to SFG rickettsiae in sentinel animals, a high environmental infestation with *A. sculptum* and the presence of capybaras (amplifying hosts of *R. rickettsii*) are important factors for BSF endemicity in Brazil. We reported these same factors in our study carried out in Goiânia, where 2 cases of rickettsioses were confirmed between 2012 and 2016 [22]. In Goiás, 16 cases of non-fatal rickettsioses have been confirmed so far [21]. Considering that mild cases of rickettsioses may be underreported, the official data must be far from the reality. Continuous surveillance and strengthening the diagnostic capabilities of public health laboratories may help to clarify the epidemiological situation of rickettsioses in Goiás and other risk areas in Brazil.

## 5. Conclusions

Our results confirmed the presence of antibodies to *Rickettsia* spp. in dogs, horses and capybaras. Most importantly, we demonstrated the exposure of dogs and horses to SFG rickettsiae. Although we only detected *R. bellii* infection in ticks, a known non-pathogenic agent for humans, this does not exclude the possible risk of rickettsiosis transmission and emphasizes the need for continuous surveillance in this area.

## Figures and Tables

**Figure 1 animals-13-01288-f001:**
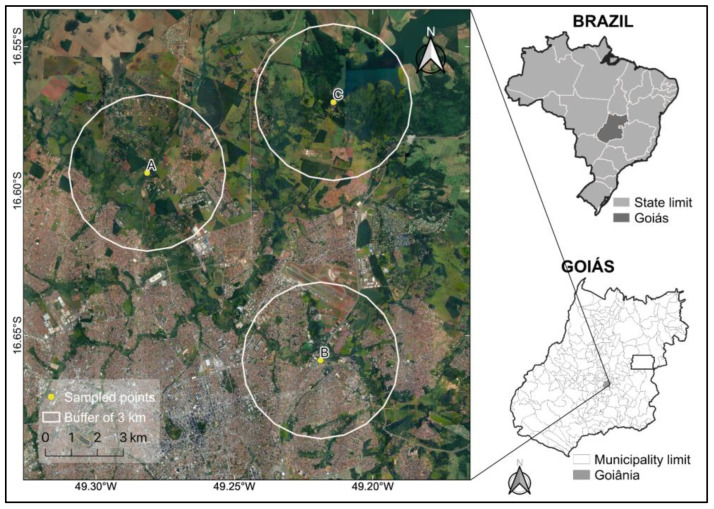
Sites where animals were sampled in the municipality of Goiânia, state of Goiás, midwestern, Brazil. A = school farm of the Veterinary and Animal Science School (EVZ) of the Federal University of Goiás and to the neighborhoods close to the EVZ; B = Vila Morais neighborhood and surrounding area; C = peri-urban and rural area close to the dam of the Ribeirão João Leite reservoir.

**Table 1 animals-13-01288-t001:** Seroreactivity for four species of *Rickettsia* from animals sampled in the municipality of Goiânia, state of Goiás, midwestern Brazil, between March 2020 and April 2022.

Animal Species and Sites	No. of Reactive/No. of Tested Samples (% Positivity) *	No. of Reactive Animals for Each *Rickettsia* Species (% Positivity for Each Animal Species/Range of Endpoint Titres)				No. of Possible Homologous Reactions Determined (PAIHR)
		*Rickettsia rickettsii*	*Rickettsia parkeri*	*Rickettsia bellii*	*Rickettsia amblyommatis*	
DOGS						
Site A (PSI)	12/50 (24.0) ^a^	5 (10.0/64–256)	3 (6.0/256–1024)	8 (16.0/128–8192)	2 (4.0/128–256)	1 (RR), 2 (RP), 8 (RB), 1 (RA)
Site B	15/59 (25.4) ^a^	8 (13.5/64–256)	3 (5.1/128)	9 (15.2/64–2048)	3 (5.1/64)	2 (RR), 6 (RB)
Site C	15/56 (26.8) ^a^	9 (16.1/64–256)	8 (14.3/64–1024)	7 (12.5/64–1024)	4 (7.1/64–128)	2 (RR), 2 (RP), 6 (RB)
**Total**	42/165 (25.4)	22 (13.3/64–256)	14 (8.5/64–1024)	24 (14.5/64–8192)	9 (5.4/64–256)	5 (RR), 4 (RP), 20 (RB), 1 (RA)
HORSES						
Site A	9/22 (40.9) ^b^	6 (27.3/64–256)	0	3 (13.6/64–256)	3 (13.6/64)	3 (RR), 1 (RB)
Site B	1/11 (9.1) ^b,c^	1 (9.1/512)	1 (9.1/64)	0	0	1 (RR)
Site C	0/11 (0) ^c^	0	0	0	0	0
**Total**	10/44 (22.7)	7 (15.9/64–512)	1 (2.3/64)	3 (6.8/64–256)	3 (6.8/64)	4 (RR), 1 (RB)
CAPYBARAS						
Site A	7/17 (41.2)	3 (17.6/64–128)	3 (17.6/256–512)	7 (41.1/128–512)	0	4 (RB)

No. = number of animals; PAIHR: possible antigen involved in homologous reaction; PSI: probable site of infection with confirmed cases of Brazilian spotted fever (BSF); RA = *R. amblyommatis*; RB = *R. bellii*; RP = *R. parkeri*; RR = *R. rickettsii*. * Seroreactivity values followed by different superscript letters represent a significant difference in the seroreactivity of the animals between the sampled areas (*p* < 0.05).

**Table 2 animals-13-01288-t002:** Number and species of ticks collected from dogs, horses, capybaras and vegetation, in the city of Goiânia, state of Goiás, midwestern Brazil, between March 2020 and April 2022.

Host/Vegetation and Sites	No. of Animals with Ticks/No. of Sampled Animals (% Infested Animals)	No. of Ticks by Species							
		*A.* *sculptum*	*A.* *dubitatum*	*A. rotundatum*	*A. ovale*	*Amblyomma* spp.	*D. nitens*	*R. sanguineus* s.l.	*R. microplus*
Dogs									
Site A	30/50 (60.0)	32 N	-	-	-	-	-	74 M, 60 F	-
Site B	26/59 (44.1)	15 N	-	-	-	-	-	128 M, 86 F	-
Site C	34/56 (60.7)	2 N	-	-	1 F	-	-	123 M, 99 F	-
Horses									
Site A	18/22 (81.8)	19 M, 14 F, 1 N	-	-	-	-	158 M, 126 F	-	-
Site B	8/11 (72.7)	6 M, 16 F	-	-	-	1 L	2 M, 16 F	-	4 M, 4 F
Site C	5/11 (45.4)	31 M, 13 F	-	-	-	-	12 F	-	2 M
Capybaras									
Site A	17/17 (100)	121 M, 79 F, 456 N	39 M, 44 F, 45 N	-	-	1 L	-	-	-
Vegetation									
Site A	n.a.	87 M, 63 F, 591 N	1 M, 1 N	-	-	1.052 L	-	-	-
Site B	n.a.	2 M, 2 F, 272 N	-	-	-	1.658 L	-	-	-
Site C	n.a.	18 M, 42 F, 180 N	1 F	1 F	-	3 L	-	-	-

M: males; F: females; N: nymphs; L: larvae. (-): absence of ixodid; No. of sites with ticks/No. of sites sampled (% infested sites). n.a.: not applicable, because it refers to ticks collected from vegetation, not from animals; in this case, in each of the sites A, B and C, ticks were collected by flagging on the vegetation in a single location.

**Table 3 animals-13-01288-t003:** Number and species of PCR-positive ticks collected from animals and vegetation in the city of Goiânia, state of Goiás, Brazil, between March 2020 and April 2022.

Host/Vegetation	Sites	No. of PCR-Positive Ticks/No. of Samples (% of Positives)
Dogs		
	Site A	0/25 (0) *R. sanguineus* s.l.
	Site B	0/12 (0) *R. sanguineus* s.l.
	Site C	0/16 (0) *R. sanguineus* s.l.
Horses		
	Site A	0/29 (0) *A. sculptum*; 0/40 (0) *D. nitens*
	Site B	0/21 (0) *A. sculptum*; 0/17 (0) *D. nitens*
	Site C	0/22 (0) *A. sculptum*; 0/10 (0) *D. nitens*
Capybaras		
	Site A	0/98 (0) *A. sculptum*; 2/26 (7.7) *A. dubitatum*
Vegetation		
	Site A	0/85 (0) *A. sculptum*; 0/1 (0) *A. dubitatum*
	Site B	0/36 (0) *A. sculptum*
	Site C	0/86 (0) *A. sculptum*

No. = number

## Data Availability

The data generated or analyzed during this study are included in this published article. The consensus sequence generated in this study is available in GenBank (OP718791).

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
