# Peer review of "Detection of Rickettsia spp. in Animals and Ticks in Midwestern Brazil, Where Human Cases of Rickettsiosis Were Reported"

_animals, 2023, doi:10.3390/ani13081288_

Round 1

Reviewer 1 Report

This is a well-conducted study attempting to find the ticks and vertebrates that play a role in human BSF in a defined region of Brazil. Despite looking hard, the answer was not really forthcoming, due to no fault of the authors.

My suggestions for improvement of the manuscript are as follows:

1. line 218. If a difference is not statistically different then one cannot call one result "higher" or "lower", than another result; it is "the same " result!

2. It is not possible to be certain of the species of Rickettsia involved in a particular infection or exposure by serology alone. Serology is not such a discriminating tool (unlike PCR or culture). In Table 2, "PAIHR"-probable antigen involved in homologous reaction-is misleading as it seems to suggest you have determined the species of Rickettsia that has stimulated the antibodies detected in that animal. This you cannot do. These conclusions need to be softened. Maybe the actual IFA titre against the 3 rickettsial antigens used in the serology should be stated for those cases where "RR" is considered to be the causative rickettsia. Then the reader can make his/her own judgement about whether it is a homologous or heterologous reaction.

3. There is very little useful information in Table 3. I would exclude it and just include the results in the text.

4. The conclusion needs to be toned down. Although antibodies to Rickettsia spp were detected in dogs, horses and capybaras, there is no certainty that they were stimulated by R.rickettsii and so the role of these 3 vertebrate species in BSF, in this area of Brazil, based on the data in the paper, is still an open question. There are plenty of SFG rickettsiae that are not human pathogens and they are likely to have stimulated these antibodies. The only definite rickettsial species that was detected (in ticks) was R.bellii, a known non-pathogen.

Author Response

Dear Reviewer 1, 

Best regards,

Reviewer 2 Report

Line 126–129: How is it possible to separate the serum from EDTA-mixed blood? Blood must be collected in serum gel and clot activator tubes to separate the serum. Explain your procedure with appropriate references and a complete procedure (like which centrifuge was used, after how long you centrifuged the sample, where you stored the collected sample, etc.).

Lines 166–169: Why molecular detection of Rickettsia spp. from the blood of dogs and horses was performed

Line 163. What is the purpose of selecting 524 tick species for DNA extraction? What is the ratio of ticks in dogs, horses, capybaras, and the environment?

Line 113. The resulting calculated sample size was 138 dogs. Then why did you screen 165 dogs? (Table 1).

Fill in the blanks in Table 2 for vegetation. Number of positive sites/number of sites examined (percentage of positive sites)

The unit of measurement should be mL instead of ml. Check and correct throughout the text.

Author Response

Dear Reviewer 2, 

Best regards,

Round 2

Reviewer 1 Report

Your paper reads much better now.

Please add a Footnote to Table 3 (having decided to retain it ) explaining "No".

line 73. replace "incipient" with "limited"

line 116. replace "in" with "to"

line 140. replace "not" with "no"

Reviewer 2 Report

Lines 126–129: The reference is missing; add a suitable reference.

Line 147: Horta et al. used serum instead of plasma. Add a suitable reference describing how plasma was used for the detection of antibodies.

Lines 166–169: Why molecular detection of Rickettsia spp. from the blood of dogs and horses was not performed (add justification in line 172).

Line 170: "Blood samples collected from capybaras (how many blood samples from capybaras were processed)"

Line 439: It is Horta, M. C.; instead of Horta, M.; (double check and correct all references)

Heading 2.6: How do you statistically analyze the DNA sample of ticks and capybaras?

No is misspelled in all three tables (1, 2, and 3).

Line 131: How many locations in the each site were screened for the collection of ticks on the vegetation? 

In Table 2, instead of writing n.a., write a specific number and percentage.

Author Response

Dear Reviewer 2, 

Best regards,

Round 3

Reviewer 2 Report

Accept in present form